# *KRAS* Exon 2 Mutations in Patients with Sporadic Colorectal Cancer: Prevalence Variations in Mexican and Latin American Populations

**DOI:** 10.3390/cancers16132323

**Published:** 2024-06-25

**Authors:** José Luis Venegas-Rodríguez, Jesús Arturo Hernández-Sandoval, Melva Gutiérrez-Angulo, José Miguel Moreno-Ortiz, Anahí González-Mercado, Jorge Peregrina-Sandoval, Helen Haydee Fernanda Ramírez-Plascencia, Beatriz Armida Flores-López, Carlos Rogelio Alvizo-Rodríguez, Jesús Alonso Valenzuela-Pérez, Sergio Cervantes-Ortiz, María de la Luz Ayala-Madrigal

**Affiliations:** 1Instituto de Genética Humana “Dr. Enrique Corona Rivera”, Centro Universitario de Ciencias de la Salud, Universidad de Guadalajara, Guadalajara 44340, Jalisco, Mexico; jluis.venegas@alumnos.udg.mx (J.L.V.-R.); qfb_arturohernandez@hotmail.com (J.A.H.-S.); melva.gutierrez@academicos.udg.mx (M.G.-A.); miguel.moreno@academicos.udg.mx (J.M.M.-O.); anahi.gonzalez@academicos.udg.mx (A.G.-M.); 2Programa de Doctorado en Genética Humana, Centro Universitario de Ciencias de la Salud, Universidad de Guadalajara, Guadalajara 44340, Jalisco, Mexico; 3Departamento de Ciencias de la Salud, Centro Universitario de los Altos, Universidad de Guadalajara, Tepatitlán de Morelos 47600, Jalisco, Mexico; 4Departamento de Biología Celular y Molecular, Centro Universitario de Ciencias Biológicas y Agropecuarias, Universidad de Guadalajara, Zapopan 44600, Jalisco, Mexico; jorge.peregrina@academicos.udg.mx; 5Facultad de Medicina, Decanato de Ciencia de la Salud, Universidad Autónoma de Guadalajara, Zapopan 45129, Jalisco, Mexico; helen.ramirez@edu.uag.mx; 6Departamento de Ciclo de Vida, Facultad de Medicina, Universidad Autónoma de Guadalajara, Zapopan 45129, Jalisco, Mexico; beatriz.flores@edu.uag.mx; 7Facultad de Medicina, Benemérita Universidad Autónoma de Puebla, Puebla 72420, Puebla, Mexico; alvrod.cr@gmail.com; 8Servicio de Colon y Recto, Hospital Civil “Dr. Juan I. Menchaca”, Guadalajara 44340, Jalisco, Mexico; dr_jvalenzuela@hotmail.com (J.A.V.-P.); sergio.cortiz@academicos.udg.mx (S.C.-O.)

**Keywords:** colorectal cancer, *KRAS* gene, exon 2 mutation, Latin American genetic variations

## Abstract

**Simple Summary:**

*KRAS* is one of the most prominent driver genes implicated in colorectal cancer (CRC), with mutations detected in 33% to 50% of CRC patients. Exon 2 harbors up to 98% of these mutations. Variants in this gene play crucial roles in the progression of the disease, influencing its development, clinical manifestations, and treatment election. This study elucidates a 17% prevalence of mutations in *KRAS* exon 2 among western Mexican patients with sporadic CRC. Furthermore, a 30% pooled prevalence of mutations in *KRAS* exon 2 was determined after analyzing an additional 16 studies from Latin America, encompassing 12,604 CRC patients. Due to advances in precision medicine treatments, knowing the pathogenic status of the *KRAS* gene will become imperative to optimally select targeted therapies.

**Abstract:**

We searched for the prevalence of actionable somatic mutations in exon 2 of the *KRAS* gene in western Mexican patients with CRC. Tumor tissue DNA samples from 150 patients with sporadic CRC recruited at the Civil Hospital of Guadalajara were analyzed. Mutations in exon 2 of the *KRAS* gene were identified using Sanger sequencing, and the data were analyzed considering clinical–pathological characteristics. Variants in codon 12 (rs121913529 G>A, G>C, and G>T) and codon 13 (rs112445441 G>A) were detected in 26 patients (with a prevalence of 17%). No significant associations were found between these variants and clinical–pathological characteristics (*p* > 0.05). Furthermore, a comprehensive search was carried out in PubMed/NCBI and Google for the prevalence of *KRAS* exon 2 mutations in Latin American populations. The 17 studies included 12,604 CRC patients, with an overall prevalence of 30% (95% CI = 0.26–0.35), although the prevalence ranged from 13 to 43% across the different data sources. Determining the variation and frequency of *KRAS* alleles in CRC patients will enhance their potential to receive targeted treatments and contribute to the understanding of the genomic profile of CRC.

## 1. Introduction

Colorectal cancer (CRC) is the third most common cancer worldwide, after breast and prostate cancers, but it ranks first in terms of deaths in Mexico [1]. As a multifactorial disease, CRC implicates environmental, epigenetic, and genetic factors [2,3]. The *KRAS* gene is one of the most involved oncogenes in signaling pathways related to CRC, particularly the MAPK pathway and other effectors, due to its impact on processes such as cell polarization, adhesion, barrier integrity maintenance, regeneration, epithelial junctions, hypoxia response, glycolysis increases, and cell proliferation control [4,5,6]. Therefore, assessing activating mutations in the *KRAS* gene that alter these processes and then detecting these mutations in CRC patients is transformative for predicting resistance to targeted therapies [7,8]. Although this gene consists of six exons, only exons 2–5 encode the MANE transcript (ID ENST00000311936.8), and exon 2 contains up to 90% of the mutations described in the gene [9,10]. Mutations located in codons 12 and 13 are multiallelic, but the changes c.35G>A and c.38G>A (p.Gly12Asp and p.Gly13Asp, respectively) are the most common [11]. Additionally, the frequency of these mutations varies between populations in CRC patients, with an overall frequency of 33–50% [12,13,14]. Besides the molecular variability in the *KRAS* gene, CRC also has a versatile clinical presentation. The presence of p.Gly12Asp and p.Gly13Asp has been associated with patients with right and left colon tumors, respectively [15]. Although CRC diagnosis is most prevalent among individuals aged 65 and older, its incidence rate has significantly risen over the past decade, with an annual increase of nearly 2% observed among individuals under the age of 50 [16]. Most patients are diagnosed after tumor stage II on initial diagnosis [17,18]. Given the significant implications of *KRAS* variants for CRC patients’ clinical features and treatment outcomes, this study aimed to assess the prevalence of exon 2 mutations in the *KRAS* gene among CRC patients from western Mexico and determine their general prevalence across Latin American populations.

## 2. Materials and Methods

### 2.1. Patients 

We enrolled 150 patients who received treatment between 2011 and 2019 at the Colon and Rectal Service of the “Dr. Juan I. Menchaca” Hospital Civil de Guadalajara. The patients were from the western region of Mexico, encompassing the states of Jalisco, Nayarit, Colima, and Michoacan. The patients meeting the inclusion criteria for sporadic colon or rectal adenocarcinoma diagnosis provided informed consent by signing the appropriate documentation for a study approved by the Research, Biosecurity, and Ethical Committees (CI-02122). The exclusion criterion was patients with non-primary tumors.

### 2.2. Methods

After obtaining tumoral tissues, DNA extraction was performed with the High Pure PCR Template Preparation Kit (Roche Diagnostics GmbH, Manheim, Germany) and quantified with a nanodrop spectrophotometer. PCR amplification of *KRAS* exon 2 was carried out with the VeritiPro Thermal Cycler using the forward primer 5′-GGCCTGCTGAAAATGACTGA-3′ and reverse primer 5′-TTACTGGTGCAGGACCATTC-3′ under the following conditions: initial denaturation at 94 °C for 5 min, followed by 29 cycles at 94 °C for 30 s, 65.4 °C for 30 s, 72 °C for 30 s, and a final extension at 72 °C for 10 min. The resulting 122 bp amplicons were separated using 2% agarose gel electrophoresis. To elucidate the possible mutations of exon 2 in the *KRAS* gene, direct Sanger sequencing was conducted using Exosap It for the purification reaction with a cycle of 37 °C for 15 min, 80 °C for 15 min, and 4 °C for 1 min. For the sequencing reaction, 5 µL of the purified PCR product was used for the BigDye Terminator V3.1 under the following PCR conditions: 96 °C for 4 min, 25 cycles at 96 °C for 10 s, 57 °C for 5 s, 60 °C for 4 min, 60 °C for 7 min, and 4 °C for 1 min. Lastly, the fragments were sequenced with a SeqStudio genetic analyzer (Applied Biosystem, Waltham, MA, USA). Chromatograms were visualized with the Chromas software V2.6.6 (Technelysium Pty Ltd., Brisbane, QLD, Australia).

To obtain the Latin American population frequencies, a systematic search was conducted in PubMed/NCBI and Google using the keywords “colorectal”, “cancer”, and “KRAS”, and the name of each of the 19 countries described in the Latin American Network Information Center of the University of Texas Libraries. Cuba, the Dominican Republic, and Puerto Rico were included despite, strictly, being Caribbean countries. The keywords and country names in Spanish were also used.

### 2.3. Statistical Analysis

The *KRAS* exon 2 mutation status for each western Mexican patient allowed the inclusion of comparison groups to assess the risk associated with clinical–pathological characteristics in wild-type individuals versus mutation carriers. Odds ratios (ORs) and confidence intervals (CIs) were used for the comparisons without adjustments. The data were analyzed using the SPSS Statistics V28.0 software (IBM Corp, Armonk, NY, USA). A significance level of *p* < 0.05 was used to determine statistical significance. Additionally, we obtained proportions for this study and others in Latin American populations to estimate a comprehensive prevalence of *KRAS* mutations for these populations. These proportions were used to compute the effect size and calculate the sampling variance. We applied a random effects model to calculate the effect size to draw more generalizable conclusions. The random effects model was estimated using the restricted maximum likelihood (REML) estimator. The observed proportion data were transformed using the Freeman–Tukey double arcsine transformation method to ensure adherence to a normal distribution. Heterogeneity was assessed using the chi-square test (Q-statistics) and the inconsistency statistic (I^2^). The heterogeneity was low if the *p*-value was greater than or equal to 0.10 and the I^2^ value was less than or equal to 50% [19]. Statistical analyses for prevalence comparisons were conducted using the R programming language with the meta package, version 4.3.1.

## 3. Results

### 3.1. Allele Frequency for KRAS Exon 2 Mutations in 150 CRC Patients from Western Mexico

Direct sequencing of 150 tumor samples was conducted to identify the prevalence of *KRAS* exon 2 mutations among CRC patients from western Mexico. The mutational analysis identified 26 patients with heterozygous mutations, indicating a 17% *KRAS* exon 2 mutation prevalence. Table 1 describes the *KRAS* variations and allele frequencies for western Mexico.

### 3.2. Comparison of Patients’ Clinical–Pathological Features

Additionally, Table 2 describes the association analysis of *KRAS* mutation and clinical–pathological features found in CRC patients from western Mexico.

### 3.3. Prevalence of KRAS Exon 2 Mutations in Latin America

In total, 126 publications were identified in the systematic search of *KRAS* exon 2 mutation frequencies in CRC patients from Latin America, of which 67 were discarded due to duplications. Of the remaining 59 publications, 16 were selected based on the following criteria: using sequencing or real-time PCR technologies, including a description of at least the number of analyzed patients and the *KRAS* exon 2 mutation frequency, and not restricting patient age.

The 16 selected studies from Latin America ranged from 2009 to 2022 and included Brazil, Chile, Colombia, Mexico, Panama, Paraguay, Peru, Puerto Rico, and Venezuela. A total of 12,604 CRC patients were analyzed, with 4112 of them being positive for a mutated allele of *KRAS* exon 2. According to the publications, the prevalence range established was 13–43%, but the overall pooled prevalence was 30% (95% CI = 0.26–0.35) (Figure 1). The analysis, stratified by methodology, revealed that 11,525 patients were examined via direct sequencing and 1079 via PCR technologies. These two subgroups exhibited an identical prevalence of 30%, with corresponding confidence intervals of 95% CI = 0.25–0.36 (*p* < 0.01, I^2^ = 84%) for direct sequencing and 95% CI = 0.22–0.38 (*p* < 0.01, I^2^ = 78%) for PCR technologies.

Pooled prevalence values of 22% for the codon 12 mutation (95% CI = 0.18–0.28; *p* < 0.01; I^2^ = 86%) and 7% for the codon 13 mutation (95% CI = 0.06–0.08; *p* < 0.01; I^2^ = 70%) were observed among the 12,284 tested patients. The 320 Peruvians studied by Aldecoa et al. [20] were not included in the prevalence analysis by mutated codon due to a lack of reported results differentiating between codon 12 and codon 13 mutations.

## 4. Discussion

*KRAS* is one of the principal driver tumor genes described in CRC and most of the deadliest types of cancer, such as pancreatic and lung adenocarcinomas. Most actionable pathogenic variants in this gene are localized in exon 2, specifically affecting codons 12 and 13 [36].

In this study, we found a mutation prevalence of 17% in exon 2 of the *KRAS* gene among 150 CRC patients from western Mexico. This observed prevalence significantly differs (*p* < 0.001) from findings reported in other Mexican cohorts. Specifically, Cárdenas-Ramos et al. documented a mutation prevalence of 34.9% [23], and Sanchez-Ibarra et al. observed a prevalence rate of 45% [25]. Neither of these studies included CRC patient populations from western Mexico. The *KRAS* exon 2 mutation prevalence of 17% is also significantly lower than the prevalence of approximately 40% observed in CRC patients across numerous studies worldwide [37,38,39]. The differences in the frequency of *KRAS* mutations could be related to racial features, as described by Staudacher et al. in a meta-analysis of 4648 CRC patients, where non-Hispanic White people exhibited fewer mutations in *KRAS* codon 12 or codon 13 than African Americans (OR 0.640; 95% CI: 0.5342–0.7666; *p* = 0.0001) [40]. Lifestyle factors may also influence somatic mutation variations for the *KRAS* gene, possibly more than the genetic structures of populations. A study found that an alkylating signature in CRC patients extensively questioned about their dietary habits before diagnosis was associated with an increased intake of red meat and the presence of *KRAS* variants such as p.G12D, p.G13D, or *PIK3CA* p.E545K in CRC tumors [41]. Additionally, an association between the *KRAS* p.Gly12Cys variant and lung adenocarcinoma in smokers has been documented [42,43,44].

Considering the multifactorial nature of CRC, involving both environmental factors and genetic predispositions among populations, investigating genetic factors in populations that exhibit maximal homogeneity should be encouraged. Therefore, we assessed the prevalence of exon 2 variants in the *KRAS* gene among Latin American populations to establish a consistent prevalence among populations with closely related genetic backgrounds. Seventeen studies, including this study, were incorporated, involving a total of 12,604 CRC patients. The pooled prevalence was 30% (95% CI = 0.26–0.35), with significant heterogeneity observed between the studies (*p* < 0.01; I^2^ = 82%).

Based on the heterogeneity findings, the Latin American population studies were further examined and stratified into two subgroups according to the mutation identification methods used: Sanger sequencing and PCR technologies. Both subgroups exhibited a consistent prevalence of 30%, with heterogeneity persisting at a consistently high level. However, the heterogeneity observed in studies utilizing PCR technologies was slightly reduced. These results suggest that the methodology employed does not significantly affect the prevalence of variants or substantially contribute to the variability observed between studies. The observed frequency variability among the studies might be more accurately ascribed to the differential tumor cell concentrations in the analyzed samples or biopsies. Furthermore, it could be influenced by extrinsic environmental factors and the intrinsic genetic heterogeneity characteristics of the populations studied, as has been previously described [40,45].

The etiology of the predominance of specific base changes resulting in amino acid alterations in *KRAS* for different cancers is not fully understood. However, variations may be attributable to the cancer’s cellular origin, the differential stress experienced by each tissue type, and the distinct impacts of risk factors associated with each carcinogenic process [42]. The principal modifications detected in cancer types characterized by frequent *KRAS* gene mutations predominantly involve the substitutions of guanine with adenine in codon 12 (c.35G>A, p.Gly12Asp) in pancreatic and colorectal adenocarcinomas and guanine with thymine (c.34G>T, p.Gly12Cys) in lung adenocarcinoma [46,47]. We found a frequency of 46% for p.Gly12Asp among the *KRAS* variants identified in CRC patients from western Mexico. This was the most frequent allele in all the selected studies from Latin American countries and most other countries. We observed frequencies of 15% and 4% for the low-frequency alleles p.Gly12Val (c.35G>T) and p.Gly12Ala (c.35G>C), respectively. These and other rare alleles, such as changes in c.34 for codon 12 and c.37 for codon 13, were observed among the Latin American studies.

The importance of identifying pathogenic variants in the *KRAS* gene is related to the application of the monoclonal antibodies cetuximab and panitumumab for patients with metastatic CRC and *RAS*/*RAF* wild-type tumors. However, targeted therapies based on *KRAS* variants have also been achieved in early-phase trials in advanced or metastatic CRC patients, with adagrasib [48] and sotorasib [49] being the first FDA-approved inhibitors that selectively target cells harboring the *KRAS* p.Gly12Cys variant. Although this is a low-frequency variant found in 3–8% of CRC patients [50,51,52], the benefits for carrier patients will be considerable. Toward this end, Wang et al. reported MRTX1133 as an inhibitor for the most frequent variant, *KRAS* p.Gly12Asp [53]. This inhibitor is being analyzed in a phase I trial for advanced solid tumors that includes CRC patients, with the study’s start date in March 2023 and the estimated completion date in August 2026 [54]. No therapies concerning the mutation of codon 13 have been described. This could be correlated with its lower prevalence distribution in colorectal tumors, possibly due to clones harboring mutations in codon 12 being more prone to the development of *KRAS*-driven cancer, thus leading to their broader distribution. The observed prevalence in Latin America may support this, where the ratio of mutations in codon 12 to those in codon 13 was 3:1. In accordance, Ahn et al. reported that mutations in *KRAS* codon 13 generally exhibit lower aggressiveness and are less likely to function as adverse prognostic markers for CRC than mutations in codon 12 [37].

The primary limitations of this study are the sample size and the patients’ origins, generally from a single institution in most of the included studies from Latin America focusing on determining the *KRAS* mutation prevalence. This is closely linked to the predominance of middle- and low-income countries in Latin America, where economic factors and the variability in health infrastructures significantly impact the availability and capacity of molecular diagnostic services, which tend to be centralized in certain governmental or private institutions. This situation results in research outcomes being derived from only a few countries. Consequently, our findings regarding the pooled prevalence of *KRAS* exon 2 mutations in Latin America may not accurately reflect the situation in countries with lower income levels, where molecular diagnostic data are scarce.

## 5. Conclusions

Mutations within the *KRAS* exon 2 region were detected in 17% of the 150 CRC patients sampled from the western Mexican population, revealing a distinctively lower incidence than the global prevalence of approximately 40%. Furthermore, the comprehensive analysis spanning Latin American studies unveiled a pooled prevalence rate of 30%, albeit accompanied by substantial heterogeneity, which could not be attributed to the employed mutation identification methods. This variability may underscore the intricate interplay between genetic and environmental factors shaping CRC susceptibility and progression within Latin American populations. In addition, these frequency disparities highlight the necessity of further research endeavors employing more homogeneous patient cohorts. Such initiatives are indispensable for elucidating the multifaceted genetic architecture of CRC across diverse populations.

Moreover, examining population-specific genetic predispositions alongside environmental factors offers significant potential for crafting targeted prevention and treatment strategies. Specifically, analyzing the exon 2 status of the *KRAS* gene can help shape treatment approaches, thereby enhancing clinical outcomes and propelling advancements in personalized medicine for the management of CRC.

## Figures and Tables

**Figure 1 cancers-16-02323-f001:**
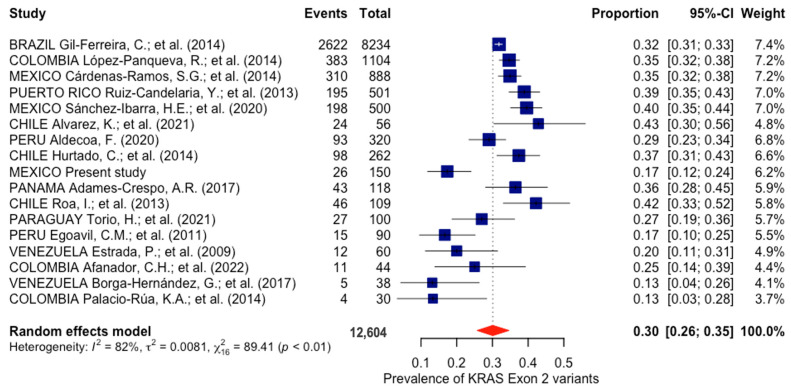
Forest plot showing the prevalence of *KRAS* exon 2 mutations in Latin America. The horizontal lines indicate the 95% confidence intervals associated with each study [20,21,22,23,24,25,26,27,28,29,30,31,32,33,34,35].

**Table 1 cancers-16-02323-t001:** Allele frequency for *KRAS* exon 2 mutations in 150 CRC patients from western Mexico.

Reference SNP	Mutation	Allele Number/Frequency
rs121913529	c.35G>A(p.Gly12Asp)	12/0.04
	c.35G>T(p.Gly12Val)	4/0.013
	c.35G>C(p.Gly12Ala)	1/0.0033
rs112445441	c.38G>A(p.Gly13Asp)	9/0.03

The mutations are shown with the nomenclature of both the DNA sequence and the protein change.

**Table 2 cancers-16-02323-t002:** Comparison of clinical–pathological features of 150 patients from western Mexico related to *KRAS* exon 2 status.

Features	*KRAS* wt	*KRAS* mut	*p*-Value *	OR (CI)
	*n* = 124	*n* = 26		
Sex				
Female	55 (44%)	9 (35%)	0.202	0.558
Male	69 (56%)	17 (65%)		(0.226–1.378)
Age				
<50	26 (21%)	9 (35%)	0.13	0.495
≥50	98 (79%)	17 (65%)		(0.197–1.243)
Tumor Localization				
Colon	76 (61%)	12 (46%)	0.218	1.71
Rectum	48 (39%)	13 (50%)		(0.723–4.06)
ND		1 (4%)		
Histological grade				
Poorly differentiated	28 (23%)	4 (15%)	0.482	1.5
Well + moderately differentiated	92 (74%)	22 (85%)		(0.477–4.77)
ND	4 (3%)			
Tumor stage				
I–II	41 (33%)	9 (35%)	0.819	0.899
III–IV	76 (61%)	15 (58%)		(0.362–2.23)
ND	7 (6%)	2 (7%)		

OR, odds ratio; CI, confidence interval; wt, wild type; mut, mutated; ND, no data. * Statistical tests: Pearson’s chi-squared test or Fisher’s exact test.

## Data Availability

The data featured in this study are accessible upon request from the corresponding author. They are not publicly available due to confidentiality agreements with this study’s participants.

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
