# Peer review of "KRAS Exon 2 Mutations in Patients with Sporadic Colorectal Cancer: Prevalence Variations in Mexican and Latin American Populations"

_cancers, 2024, doi:10.3390/cancers16132323_

Round 1

Reviewer 1 Report

Comments and Suggestions for Authors

The article is written in an appropriate way. Data and analysis are presented appropriately. Methods, tools, software and reagents are described in considerable detail.

1. prevalence was assessed in Mexican and Latin-American populations, 2.  were searched somatic mutations in exon 2 for the KRAS gene in  West Mexican patients with CRC, 3. only 150 patients were examined - this is too small a study group, 4. there is no element of novelty, the research has only been repeated in a different population, 5. the conclusions are too general, 6. I have no objections to references,

7. I have no objections to the tables and figures

Author Response

Dear Reviewer, 

We are thankful for your attention.

Sincerely,

María de la Luz Ayala-Madrigal

Reviewer 2 Report

Comments and Suggestions for Authors

the Authors present a noticeable study investigating the prevalence of KRAS exon 2 mutations in CRC patients from Western Mexico, as well as to determine a general prevalence across Latin America populations.

the study is well conducted, but I personally feel that the aim is too broad. In my opinion, it would have been preferred to provide an analysis of Western Mexico in a first study and then inferring a general prevalence.

However, this is only my impression. On a more general note, can the Authors further clarify the methods used to get a generizable conclusion from the heterogeneous studies presented? this should be commented also in the Discussion I think.

Comments on the Quality of English Language

fine, only minor editing required at the proofs stage.

Author Response

Dear Reviewer, 

We appreciate your review and comments.

Sincerely,

María de la Luz Ayala-Madrigal

Reviewer 3 Report

Comments and Suggestions for Authors

In this manuscript the authors show the prevalence of the most common KRAS exon 2 mutations in sporadic colorectal cancer patients. The manuscript is generally well explained and well written and the results are quite clear. However, my biggest concern about this manuscript is the overall importance of the results the authors report. I have couple of questions that are stated bellow:

-          Please describe patient group in more detail (like histopathological diagnosis, sex, age, TNM stage, etc.) as well as inclusion and exclusion criteria.

-          Did all 150 patients come from one hospital? What area does this hospital include? Just the city or wider region?

-          Which result from this study do the authors consider as new, innovative, or groundbreaking and why?

Comments on the Quality of English Language

-          There are some language errors that should be corrected by a native English speaker.

Author Response

Dear Reviewer, 

We are thankful for your feedback.

Sincerely,

María de la Luz Ayala-Madrigal
